# Impact of Antibiotic Prophylaxis on Surgical Site Infections in Cardiac Surgery

**DOI:** 10.3390/antibiotics12010085

**Published:** 2023-01-04

**Authors:** Christian de Tymowski, Tarek Sahnoun, Sophie Provenchere, Marylou Para, Nicolas Derre, Pierre Mutuon, Xavier Duval, Nathalie Grall, Bernard Iung, Solen Kernéis, Jean-Christophe Lucet, Philippe Montravers

**Affiliations:** 1Department of Anaesthesiology and Surgical Intensive Care, DMU PARABOL, AP-HP, Hôpital Bichat, 75018 Paris, France; 2Université Paris Cité, Centre de Recherche sur l’Inflammation, INSERM UMR 1149, CNRS ERL8252, F-75018 Paris, France; 3Laboratory of Excellence, Inflamex, Université Paris Cité, F-75018 Paris, France; 4Department of Immunology, DHU Fire, AP-HP, Hôpital Bichat, 75018 Paris, France; 5INSERM Clinical Investigation Center 1425, 75018 Paris, France; 6Department of Cardiac Surgery, AP-HP, Hôpital Bichat, 75018 Paris, France; 7UFR Paris Nord, Université Paris Cité, 75006 Paris, France; 8Service MSI, AP-HP, Hôpital Bichat, 75018 Paris, France; 9Université Paris Cité, INSERM, IAME, F-75018 Paris, France; 10Service de Bactériologie, AP-HP, Hôpital Bichat, 75018 Paris, France; 11Cardiology Department, AP-HP, Bichat Hospital, Université Paris Cite, INSERM 1148, 46 Rue Henri Huchard, 75018 Paris, France; 12Equipe de Prévention du Risque Infectieux (EPRI), AP-HP, Hôpital Bichat, 75018 Paris, France; 13Université Paris Cité, Physiopathologie et Epidémiologie des Maladies Respiratoires, INSERM UMR 1152, F-75018 Paris, France

**Keywords:** antibiotic prophylaxis, vancomycin, gentamicin, cephalosporins, surgical site infection, cardiac surgery, cardiopulmonary bypass

## Abstract

(1) Background: Cephalosporins (CA) are the first-line antibiotic prophylaxis recommended to prevent surgical site infection (SSI) after cardiac surgery. The combination of vancomycin/gentamicin (VGA) might represent a good alternative, but few studies have evaluated its efficacy in SSI prevention. (2) Methods: A single-centre retrospective study was conducted over a 13-year period in all consecutive adult patients undergoing elective cardiac surgery. Patients were stratified according to the type of antibiotic prophylaxis. CA served as the first-line prophylaxis, and VGA was used as the second-line prophylaxis. The primary endpoint was SSI occurrence at 90 days, which was defined as the need for reoperation due to SSI. (3) Results: In total, 14,960 adult patients treated consecutively from 2006 to 2019 were included in this study, of whom 1774 (12%) received VGA and 540 (3.7%) developed SSI. VGA patients had higher severity with increased 90-day mortality. Nevertheless, the frequency of SSI was similar between CA and VGA patients. However, the microbiological aetiologies were different, with more Gram-negative bacteria noted in the VGA group. (4) Conclusions: VGA seems to be as effective as CA in preventing SSI.

## 1. Introduction

Since the advent of antibiotics in the 1940s [1,2], antibiotic prophylaxis has played an important role in the prevention of infections, including cardiac surgery [3]. However, 80 years later, despite international guidelines [4], the modalities of the administration of antibiotic prophylaxis remain heterogeneous [5]. In cardiac surgery, the reference prophylaxis policy directs the administration of cephalosporins before skin incision and up to 24–48 h after, and the main alternative is the administration of vancomycin [4,6]. Indeed, in many cardiac surgery studies, vancomycin appears less effective for preventing surgical site infections (SSIs) than anti-staphylococcal penicillins or first/second-generation cephalosporins, especially those infections caused by methicillin-susceptible staphylococci (MSS) [7,8,9,10]. This finding might be explained by the reduced bactericidal effect of vancomycin [11,12] and less optimal penetration into mediastinal tissues during cardiac surgery under cardiopulmonary bypass (CPB) [13]. However, several studies did not identify major differences between these two antibiotic regimens, particularly when the duration of administration was considered [14,15]. In addition, some studies have found a significant effect of the timing of vancomycin injection, with an increased risk of SSI in cases of delayed administration [15,16]. Finally, in our centre, vancomycin is combined with gentamicin to enlarge the spectrum of prophylaxis and target Gram-negative bacterial infections [17]. Interestingly, few studies have compared the efficacy of the combination of vancomycin and gentamicin to that of cephalosporins.

Thus, the aim of our retrospective, single-centre study was to assess whether the combination of vancomycin and gentamicin (VGA) is a valuable alternative to cephalosporins (CA) to prevent SSI in cardiac surgery with CPB.

## 2. Materials and Methods

### 2.1. Study Design

This single-center retrospective study was performed in a tertiary care teaching hospital performing more than 1000 annual cardiac surgical procedures. Between 2006 and 2019, all consecutive patients who underwent cardiac surgery with cardiopulmonary bypass (CPB) were eligible. The exclusion criteria were a previous history of cardiac surgery in the past year and missing clinical information (including antibiotic prophylaxis).

### 2.2. Definitions

SSI was defined as the need for reoperation within 90 days after index surgery for local or systemic infection involving the sternotomy scar [18,19]. The sternal scar was assessed daily after surgery, and the date of SSI was the date of return to the operating room for SSI. Only the first reoperation for suspected SSI was considered. The validation of SSI was made during multidisciplinary regular meetings including a cardiac surgeon, an intensive-care physician, and a physician of the infection control unit. Deep SSI (dSSI) was defined as the need for sternal reopening with deep purulent discharge or destruction of the sternal bone/deep sternal osteomyelitis. Otherwise, the SSI was defined as a superficial SSI (sSSI) requiring reoperation. Obesity was defined as a body mass index >30 kg/m^2^. Acute infective endocarditis was defined as endocarditis under antibiotics at the time of surgery. Cardiac insufficiency corresponds to patients with a left ventricular ejection fraction of less than 30%.

### 2.3. Selection of Antibiotic Prophylaxis and Protocol of Administration

During the study period, first- and second-generation cephalosporins were used the first-line intraoperative antibiotic prophylaxis. Cefamandole was administered from 2006 to 2015, and cefazolin was administered from 2016 to 2019. Cephalosporins were administered according to the following protocol: 30 mg/kg at induction with a reinjection every 4 h during the operation and then a dose of 1 g per 6 h for 24 h. Moreover, 1 g was administered into the priming of the CB. The combination of vancomycin/gentamicin was the second-line antibiotic prophylaxis selected in cases of allergy to beta-lactam, prior antibiotherapy ≥ 72 h, prior hospitalization in a unit with a high prevalence of methicillin-resistant *Staphylococcus aureus* (MRSA) and/or in patients known to be colonized by MRSA. VGA was administered after the induction and from 5 to 30 min before the surgical incision through a central line according to the following protocol: vancomycin was perfused over 1 to 2 h at a dose of 30 mg/kg, and 500 mg was added to the priming of CPB. Then, 1 g was administered 12 h after the end of surgery. Gentamicin was perfused over 30 min as a single dose of 5–6 mg/kg. All patients underwent preoperative nasal screening for *S. aureus* carriage and preoperative nasal decontamination with mupirocin for 5 days, most frequently started on the day before surgery [20].

### 2.4. Objectives

The primary outcome was SSI occurrence at D-90 after index surgery. Secondary outcomes were the type of SSI, microbiological features, death at D-90, and occurrence of SSI based on propensity score-matching.

### 2.5. Data Collection and Ethics

Data were extracted from three databases. The first registry, Cardiobase, prospectively includes all adult patients who underwent cardiac surgery with CPB. This database has been approved by the Paris-Nord Biomedical Research Project Evaluation Committee (CERB, IRB No. 00006477), the Advisory Committee on Information Processing in Health Research (Paris, France) and the Commission Nationale Informatique et Liberté (CNIL, Paris, France). It is registered on ClinicalTrials.gov (NCT03393169). The following data are collected in this registry: demographic data, medical history, type of surgery and perioperative data. The Cardiobase information on SSIs was combined with a second database collected by the Infection Control Unit [18,20]. Finally, survival data were extracted from public French data of Institut National de la Statistique et des Etudes Economiques (INSEE) from 2006 to 2021, where the date of death was recorded.

### 2.6. Patient Management

Chronic therapies were administered on the day of surgery except for angiotensin-converting enzyme inhibitors/angiotensin receptor blockers. CA was administered at the time of anaesthesia induction, whereas VGA was infused after induction when the central venous catheter was inserted and before the surgical incision. Otherwise, patient intraoperative management was performed as previously described [21]. In our centre, coronary artery bypass grafting (CABG) is performed only with the internal mammary arteries (IMA), even for multivessel disease. For this purpose, the two IMA are harvested using the “skeletonization technique” (without collateral veins or endothoracic fascia). Then, the right IMA is implanted as a free graft on the left IMA using the so-called Y or T techniques. Finally, the use of sequential distal lateral anastomoses allows for multiple coronary branches to be targeting while respecting a harmonious bypass flow path. Secondary angioplasty is reserved for rare cases where the downstream bed of the right coronary is not accessible to bypass. After surgery, patients were admitted to the cardiac surgical intensive care unit for at least 48 h.

### 2.7. Statistical Analysis

Continuous variables were expressed as medians with interquartile ranges (IQRs) and were compared with the Mann–Whitney U test or the Kruskal–Wallis test as appropriate. Categorical variables were expressed as absolute numbers and proportions and were compared with Fisher’s exact test or the chi-square test, as appropriate.

Time-to-event analyses were estimated with Kaplan–Meier analyses. To assess whether SSI was an independent factor for death, multivariate analyses were performed using a Cox model. For all models, variables with nominal two-tailed *p* values less than 0.05 were entered into the multivariate model, except for variables with obvious collinearity. The final models were selected using backwards stepwise regression and evaluated using the AIC and Tjur’s R^2^ coefficient of discrimination.

To address confounding by indication of VGA and other sources of bias arising from the use of observational data, we estimated full propensity score matching without replacement [22] using the Matchit package [23], which calls functions from the optimatch package [24]. The propensity score was estimated using logistic regression of the likelihood of total VGA administration (see Appendix A), and covariate balance was assessed using the cobalt package [25]. All statistical analyses were performed using R software (R Core Team, 2014, R Foundation for Statistical Computing, Vienna, Austria). Figures were produced using the ‘ggplot2 package’, and statistics were produced using the ‘stat package’. Missing values were not imputed. A *p* value < 0.05 was considered statistically significant. Reports of statistical analyses respected the STROBE recommendations for cohort studies [26] (checklist available in Appendix B).

## 3. Results

### 3.1. Homogeneity of the Cohort

From 2006 to 2019, 14,770 (99%) patients out of 14,960 fulfilled the inclusion criteria (Figure 1). The annual number of patients included was stable throughout the study period (Appendix A). Among these patients, 540 (3.6%) developed a postoperative SSI, 151 (1%) a dSSI, and 389 (2.6%) an sSSI requiring reoperation. Whereas SSI occurrence remained stable across time (Appendix A), a reduction in VGA use was observed from 16% in 2010–2011 to 7% in 2018–2019 (*p* < 0.001) (Appendix A).

### 3.2. Patient Characteristics Based on the Type of Antibiotic Prophylaxis

Compared to patients who received CA, those who received VGA were younger and had a lower BMI, fewer cardiovascular risk factors (arterial hypertension, diabetes mellitus, and dyslipidaemia) and a lower proportion of ischaemic heart disease (52% vs. 32%, *p* < 0.001) (Table 1). However, these patients had more comorbidities, including cardiac insufficiency (4.3 vs. 6.5%, *p* < 0.001), chronic obstructive pulmonary disease (COPD), and cirrhosis. Moreover, VGA was strongly associated with surgical emergency, preoperative critical state and acute infective endocarditis. Overall, 62% (474/759) of patients with endocarditis received VGA. Similarly, the EuroSCORE II and the CPB duration were higher in patients who received VGA (4% (2–9) vs. 2% (1–4) and 71 min (51–104) vs. 58 min (45–83), respectively, both *p* < 0.001). VGA was more frequently used in valve surgery than in CABG (17% (993/5794) vs. 5.7% (328/5715), *p* < 0.001). Finally, VGA, compared to CA, was associated with a more complicated postoperative course with more catecholamine requirements, higher blood loss, longer ICU length of stay and higher 90-day (D90) mortality (10% vs. 4.8%, *p* < 0.001, Appendix A). 

### 3.3. Relationship between Antibiotic Prophylaxis and SSI in the General Population

The type of antibiotic prophylaxis was not associated with SSI occurrence (Table 1 and Figure 2), and the time to SSI was similar between VGA and CA (15 (12–26) vs. 18 (12–27) days, *p* = 0.77). The main independent parameters associated with SSI (Table 2 and Appendix A) included older age, female sex, obesity, diabetes mellitus, cardiac insufficiency, ischaemic heart disease, and end-stage renal disease. Furthermore, SSIs were associated with CABG, short CPB duration and catecholamine requirements after CB. In multivariate analysis, VGA was not associated with SSI (Table 2).

Of note, in univariate analysis, the occurrence of sSSI appeared to be lower in the VGA group than in the CA group (Table 1 and Figure 3A), without a difference in the timing of sSSI occurrence. However, this association did not persist in multivariate analysis (Appendix A). No association was found between dSSI occurrence and type of prophylaxis.

### 3.4. Microbiological Characteristics of Patients According to the Type of SSI

The microbiological nature of the SSI was different according to the type of prophylaxis (Table 3). Gram-negative bacteria were more frequently noted in VGA than in CA (52% vs. 35%, *p* = 0.013). In contrast, Gram-positive bacteria were more frequently noted in CA (69% vs. 50%, *p* = 0.006), especially enterococci, representing 13% of SSI in the CA group. In contrast, only one enterococcal infection was observed in the VGA group.

### 3.5. Relationship between Antibiotic Prophylaxis and SSI in Propensisty Score Analsysis

Given the heterogeneity of the study population, three sensitivity analyses were performed. The first was a propensity score using the full matching method.

In this analysis, the main risk factors associated with SSI (Table 4), sSSI and dSSI (Appendix A) remained the same. However, acute infective endocarditis became a risk factor for dSSI, and cardiac transplantation was only associated with dSSI. Of note, in these analyses, end-stage renal disease (ESRD) was the risk factor with the highest OR. In all analyses, VGA was not associated with SSI.

### 3.6. Relationships between Antibiotic Prophylaxis and SSI in a Sensitivity Analysis Excluding Active Infective Endocarditis

The second sensitivity analysis excluded patients with active infective endocarditis. Indeed, these patients may have a lower risk of SSI due to concomitant antibiotic therapy. In addition, VGA was over-represented among these patients, with 62% receiving VGA, while they represented only 12% of the overall population. In this analysis, VGA patients still presented more comorbidities and severity risk factors with higher EuroSCORE II compared to CA patients (3 (2–7) vs. 2 (1–4), *p* < 0.001) (Appendix A). No difference was noted in the occurrence of SSI, sSSI and dSSI (Figure 4). In multivariate analysis, VGA was not a risk factor of SSI (Table 5), sSSI or dSSI (Appendix A).

### 3.7. Relationships between Antibiotic Prophylaxis and SSI in a Sensitivity Analysis Restricted to CABG

Finally, a sensitivity analysis was performed in the homogeneous subgroup of patients undergoing coronary artery bypass grafting, a population at high risk of developing SSI. In this subgroup, VGA was less frequently associated with patient severity (Appendix A). In the VGA group, patients were more often female, with a higher frequency of insulin-dependent diabetes, end-stage renal failure and chronic obstructive pulmonary disease, and slightly greater EuroSCORE II (2 (1–3) vs. 1 (1,2), *p* < 0.001) (Appendix A).

In this analysis, SSI were more frequent in the VGA group compared to the CA group (9.8% vs. 6.3, *p* = 0.012, Figure 5A), and especially superficial SSI (7.6% vs. 5%, *p* = 0.038, Figure 5B) but without a difference in the delay of occurrence (16 days (12–24) vs. 19 (13–27), *p* = 0.18, respectively). However, in multivariate analysis, VGA was not associated with SSI occurrence (OR = 1.27 [0.84–1.92], *p* = 0.254, Table 6), with sSSI or dSSI (Appendix A).

## 4. Discussion

### 4.1. Summary of the Main Results

In our long-term cohort, VGA was prescribed more frequently in the most severe patients, including those with acute infective endocarditis. In univariate analysis, VGA was associated with a lower risk of sSSI, but this association did not persist in multivariate analysis. In addition, no association was found with sSSI and dSSI, even when performing three sensitivity analyses. Therefore, our study suggests that VGA could represent a good alternative to cephalosporins. However, our study also underlines the microbiological differences according to the type of prophylaxis. Specifically, VGA was associated with more Gram-negative bacteria and fewer Gram-positive bacteria in SSI.

### 4.2. Comparison with Other Studies and Physiopathological Hypotheses

#### 4.2.1. SSI Occurrence According to the Type of Antibiotic Prophylaxis

The similar efficacy of CA and VGA to prevent SSI is consistent with the large multicentre retrospective study on 21,396 patients [27] and the 2012 meta-analysis, which included nine studies, regardless of the duration of antibiotic prophylaxis, five of which had a similar duration of administration of vancomycin and cephalosporins [14].

In the overall analysis of the meta-analysis that included 6022 patients, vancomycin administration was associated with a slight increase in the frequency of SSIs (OR = 1.33 [1.08–1.64]), regardless of the duration of antibiotic prophylaxis, but this association disappeared in the subgroup analysis that included 2278 patients with a similar duration of antibiotic prophylaxis (OR = 0.81 [0.57–1.15]).

#### 4.2.2. Prevention of Gram-Negative SSI by Gentamicin

In addition, our study suggests an increased incidence of Gram-negative bacteria in the VGA group. This association was not found in the 2012 meta-analysis, although this analysis found more postoperative Gram-negative pneumonia in the vancomycin group. This lack of association in this meta-analysis may be due to a lack of power, as our population was more than twice as large [14]. The increase in Gram-negative SSI is surprising and deceptive, given that gentamicin was associated with vancomycin to reduce Gram-negative bacteria infection. This finding is in contrast to that noted in the meta-analysis, where vancomycin was used alone [14]. From this perspective, gentamicin seems to have a minimal impact on the prevention of SSI. A first explanation for this could be the high prevalence of gentamicin resistance in a nosocomial environment [28,29]. However, active endocarditis patients, who are at high risk of bacterial resistance due to long term antibiotherapy, had a low prevalence of SSI. In addition, the better coverage of Gram-positive bacteria, such as enterococci, offered by vancomycin could favour Gram-negative infections. Indeed, cephalosporins are not effective against this pathogen [30,31], which may explain why, out of the 63 enterococcal SSIs in our study, 62 occurred in the CA group, compared with one in the vancomycin group. Therefore, for patients at high risk for SSI who are not allergic to beta-lactams, our results suggest that the combination of vancomycin and cefazolin could be tested to provide better coverage of Gram-negative and Gram-positive bacteria, as used by some teams [16]. A prospective multicentre study evaluating this combination would provide answers [32]. Otherwise, broad-spectrum cephalosporins, such as ceftaroline [33,34], could represent another option, providing a more protective effect for patients allergic to beta-lactam. This antibiotic offers two advantages for our combination of VGA. First, in the absence of veinotoxicity, it could be administered before central-line insertion. Second, it is less nephrotoxic, and the prophylactic administration of vancomycin and, especially, the combination of vancomycin/aminoglycoside has been associated with an increased risk of acute kidney injury [27]. However, no data are available for this indication. The results from 2014, assessing ceftaroline for this indication, are not available (clinical trial number = NCT02307006) [35], and the risk of selecting new resistance and the alteration in gut microbiota may limit its use.

#### 4.2.3. Timing of Antibiotic Prophylaxis and Role of Gentamicin

Nevertheless, for patients allergic to beta-lactams, gentamicin may reinforce the administration of vancomycin. Indeed, in our study, the exact start date of antibiotic prophylaxis was not collected. However, in our protocol, VGA started before the surgical incision and after the insertion of the central line. Therefore, our protocol is not in accordance with French and international recommendations [6], which suggest that the vancomycin infusion should end before the surgical incision. Indeed, some studies have shown an increased risk of SSI when vancomycin is delayed [15,16]. Garey et al. [15] reported that starting vancomycin 0–15 min before the surgery incision was associated with an increased risk of SSI with an OR of 11.6 (2.6–52.4) compared to patients where vancomycin was started 16–60 min before incision. Three hypotheses could explain the discrepancy with our study. First, the population of the Garey et al. study was small (15 patients in the 0- to 15-min group and 176 in the 16- to 60-min group) and may lead to an overestimation of the risk. Second, some patients began receiving vancomycin from 15 to 30 min before incision, which is the optimal time according to this study. Third, gentamicin was not used in this study. This antibiotic offered a fast bactericidal effect [36], which is increased by vancomycin [37]. Therefore, gentamicin may offer some advantages.

### 4.3. Limits and Strengths

This study is one of the largest studies on SSI in cardiac surgery and, more specifically, on the combination of vancomycin/gentamicin. The inclusion of patients and their surgical management were stable over time. Thus, the SSI rate was stable throughout the study. However, the use of VGA therapy decreased over time, which was possibly related to a decrease in the MRSA rate in our population.

Other limitations are present. First, this is a retrospective study, which may lead to collection bias. This bias is limited by the fact that much of the information was collected prospectively. Second, it is a monocentric study, which may limit generalization. This study found SSI rates and risk factors for SSI in line with previously published studies [38,39,40,41], and the results are consistent with previously published articles [14,27]. Thus, we assume that this bias is limited. Third, the exact timing of the injection of the antibiotic was not collected, which does not allow for us to take this factor into account in the occurrence of SSI. Fifth, the patients in the VGA group had a higher mortality rate than those in the CA group. Given this heterogeneity, we performed a propensity score analysis, which showed similar results. We used the “full matching” method, which has the advantage of retaining the whole population by assigning a weight to each patient. Then, we performed two additional subgroup analyses, first by excluding active endocarditis and then by selecting a homogeneous population of patients undergoing isolated CABG. The fact that the results of all three sensitivity analyses concurred with the full population analysis limits this severity bias.

## 5. Conclusions

In this prolonged cohort of consecutive patients, the incidence and risk factors for SSI were comparable to those reported in the literature. VGA was more commonly used in the most severe patients than in CA. However, our results did not find any significant increase in risk of SSI. Differences in the microbiological cultures of SSI were observed according to the type of antibiotic prophylaxis. Therefore, VGA represents a valuable alternative for SSI prevention. Further studies are needed to confirm these results.

## Figures and Tables

**Figure 1 antibiotics-12-00085-f001:**
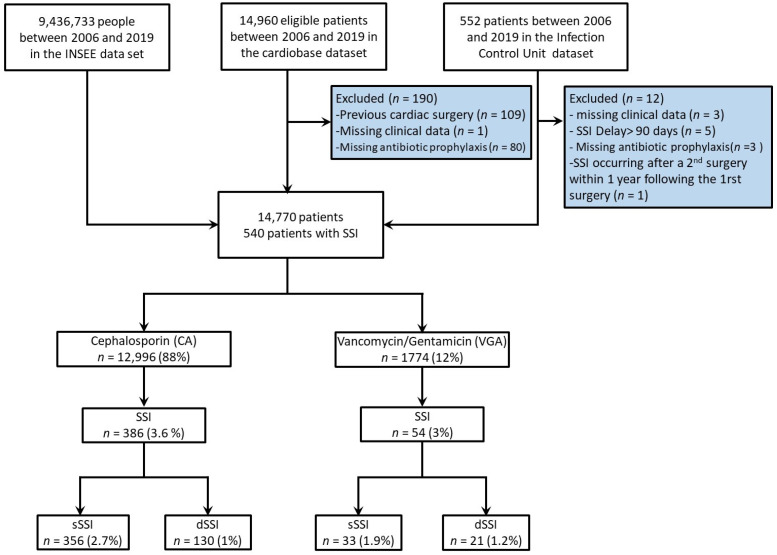
Flow chart of the study. INSEE: Institut National de la Statistique et des Etudes Economiques; SSI, surgical site infection. dSSI, deep SSI; sSSI: superficial SSI.

**Figure 2 antibiotics-12-00085-f002:**
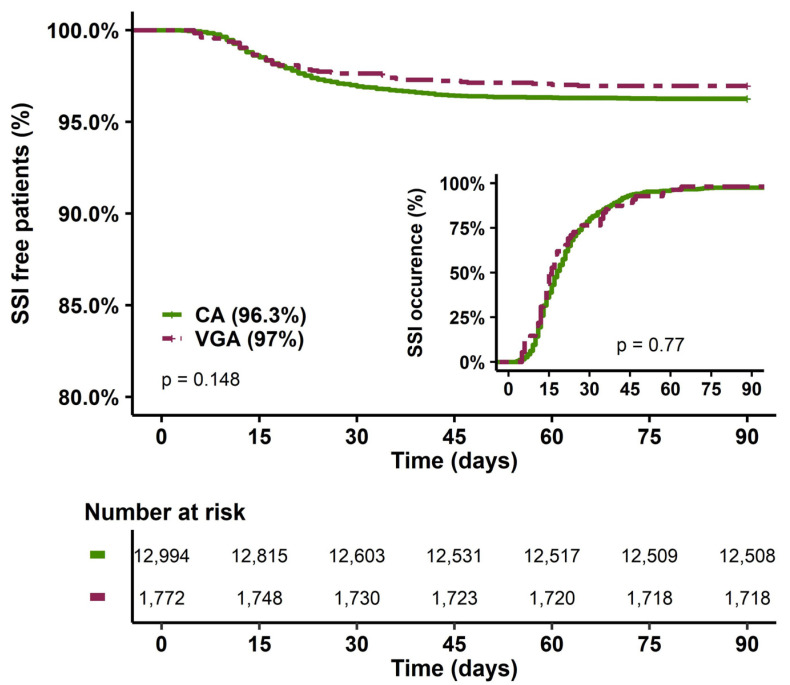
SSI-free patients at 90 days and delay of SSI occurrence according to the type of prophylaxis. CA, cephalosporin antibiotic prophylaxis; VGA, vancomycin/gentamicin antibiotic prophylaxis; SSI, surgical site infection.

**Figure 3 antibiotics-12-00085-f003:**
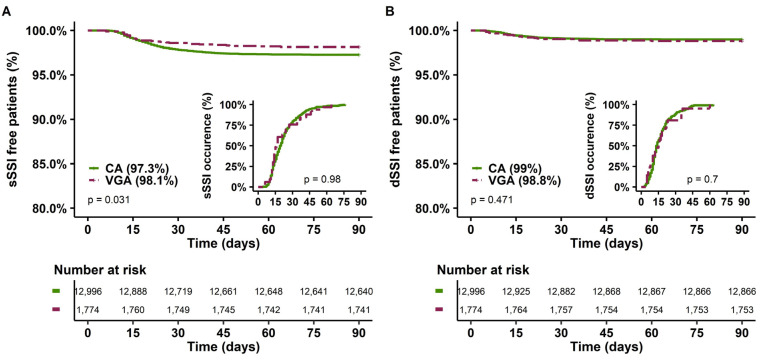
sSSI-free and dSSI-free patients at 90 days and delay of SSI occurrence according to the type of prophylaxis. (**A**) Superficial SSI; (**B**) Deep SSI. CA: cephalosporin antibiotic prophylaxis, VGA vancomycin/gentamicin antibiotic prophylaxis. SSI, surgical site infection.

**Figure 4 antibiotics-12-00085-f004:**
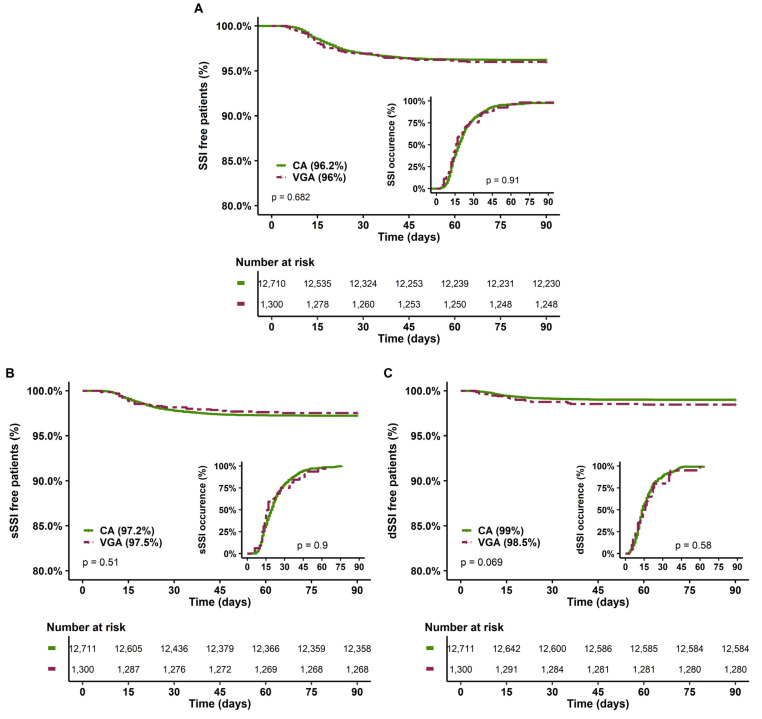
SSI-free patients at 90 days and delay of SSI occurrence according to the type of prophylaxis in the sensitivity analysis, excluding infective active endocarditis. (**A**) All SSI, (**B**) Superficial SSI; (**C**) Deep SSI. CA, cephalosporin antibiotic prophylaxis; VGA, vancomycin/gentamicin antibiotic prophylaxis; SSI, surgical site infection.

**Figure 5 antibiotics-12-00085-f005:**
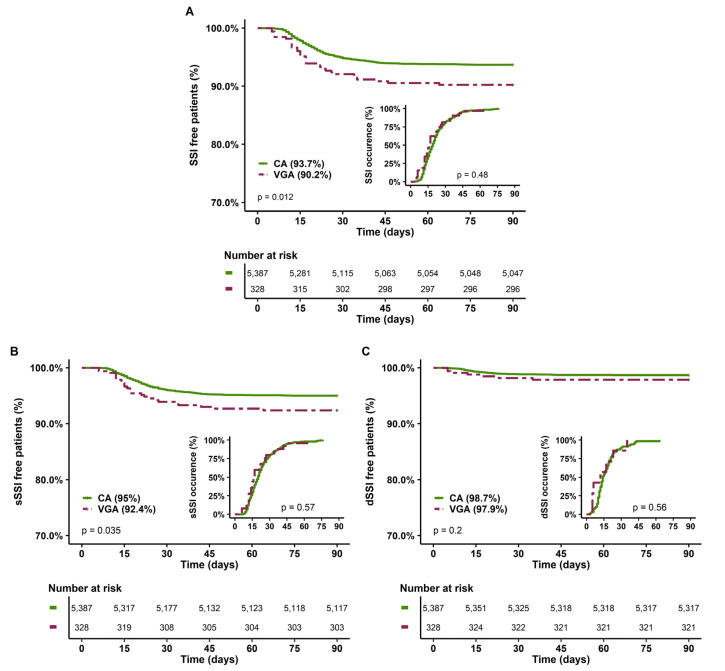
SSI-free patients at 90 days and delay of SSI occurrence according to the type of prophylaxis in the sensitivity analysis restricted to CABG. (**A**) All SSI, (**B**) Superficial SSI; (**C**) Deep SSI. CA, cephalosporin antibiotic prophylaxis; CABG, coronary artery bypass graft, VGA, vancomycin/gentamicin antibiotic prophylaxis; SSI, surgical site infection.

**Table 1 antibiotics-12-00085-t001:** Perioperative characteristics of patients according to the type of prophylaxis.

	Overall *n* = 14,770 (100%)	Missing Value	CA *n* = 12,996 (88%)	VGA *n* = 1774 (12%)	*p* Value
Demography					
Male	10,225 (69)	0	9153 (70)	1072 (60)	<0.001
Age (years)	66(56–74)	0	66 (56–74)	64 (53–74)	<0.001
1st quartile < 56 years	3692 (25)	0	3163 (24)	529 (30)	<0.001
2nd quartile (56, 65) years	3693 (25)	0	3262 (25)	431 (24)	<0.001
3rd quartile (66–74) years	3691 (25)	0	3290 (25)	401 (23)	
4th quartile > 74 years	3694 (25)	0	3281 (25)	413 (23)	
BMI (kg/m^2^)	26.2 (23.6–29.4)	101	26.2 (23.7,29.4)	25.6 (22.8–29.1)	<0.001
Obesity	3275 (22)	101	2906 (23)	369 (21)	0.127
Medical history					
Smoking	5299 (36)	0	4651 (36)	648 (37)	0.542
Arterial hypertension	8675 (59)	0	7733 (60)	942 (53)	<0.01
Diabetes mellitus	4036 (27)	0	3602 (28)	434 (24)	0.004
Insulin-dependent diabetes	1245 (8.4)	0	1086 (8.4)	159 (9.0)	0.389
Noninsulin-dependent diabetes	2787 (19)	0	2513 (19)	274 (15)	<0.001
Dyslipidaemia	7432 (50)	0	6717 (52)	715 (40)	<0.001
Chronic peripheral arterial insufficiency	1801 (12)	0	1600 (12)	201 (11)	0.236
Stroke	1333 (9.0)	0	1055 (8.1)	278 (16)	<0.001
Cardiac insufficiency	670 (4.5)	0	555 (4.3)	115 (6.5)	<0.001
Ischaemic heart disease	7388 (50)	0	6819 (52)	569 (32)	<0.001
ESRD	191 (1.3)	0	130 (1.0)	61 (3.4)	<0.001
COPD	1211 (8.2)	0	1007 (7.7)	204 (11)	<0.001
Cirrhosis	148 (1.0)	0	108 (0.8)	40 (2.3)	<0.001
Preoperative data					
B-blocker	8620 (58)	0	7853 (60)	767 (43)	<0.001
ACE inhibitor	7184 (49)	0	6459 (50)	725 (41)	<0.001
Statins	8474 (57)	0	7711 (59)	763 (43)	<0.001
Antiplatelet agent	8430 (57)	0	7691 (59)	739 (42)	<0.001
Aortic regurgitation	1472 (10.0)	0	1096 (8.4)	376 (21)	<0.001
Mitral regurgitation	1794 (12)	0	1415 (11)	379 (21)	<0.001
Aortic Stenosis	3664 (25)	0	3266 (25)	398 (22)	0.014
Mitral stenosis	1038 (7.0)	0	902 (6.9)	136 (7.7)	0.262
Surgical emergency	1100 (7.4)	0	868 (6.7)	232 (13)	<0.001
Acute infective endocarditis	759 (5.1)	0	285 (2.2)	474 (27)	<0.001
Prior cardiac surgery	1345 (9.1)	0	1001 (7.7)	344 (19)	<0.001
Preoperative critical state	542 (3.7)	0	357 (2.7)	185 (10)	<0.001
MV	259 (1.8)	0	141 (1.1)	118 (6.7)	<0.001
Catecholamine	312 (2.1)	0	200 (1.5)	112 (6.3)	<0.001
AKI	222 (1.5)	0	118 (0.9)	104 (5.9)	<0.001
Haemoglobin (g/dL)	13.4 (12.0–14.5)	317	13.5 (12.3–14.6)	12.2 (10.5–13.6)	<0.001
Platelet count (G/L)	220 (182–266)	525	219 (182–264)	226 (181–284)	<0.001
Prothrombin ratio (%)	93 (82,100)	770	94 (83–100)	88 (75–100)	<0.001
Creatinin (µg/L)	88 (75,107)	250	88 (75–106)	91 (75–120)	<0.001
EuroSCORE II	2 (1–5)	0	2 (1–4)	4 (2–9)	<0.001
Intraoperative data					
ACC time	47 (36–66)	386	46 (36, 64)	55 (39, 78)	<0.001
CBP time	59 (46, 85)	340	58 (45, 83)	71 (51, 104)	<0.001
Isolated CABG	5715 (39)	5387 (41)	5387 (41)	328 (18)	<0.001
Bimammary artery bypass	5754 (39)	0	5391 (41)	363 (20)	<0.001
CABG and valve surgery	1380 (9.3)	0	1192 (9.2)	188 (11)	0.053
Isolated valvular surgery	5794 (39)	0	4801 (37)	993 (56)	<0.001
Thoracic aortic surgery	1508 (10)	0	1334 (10)	174 (9.8)	0.552
Cardiac transplantation	214 (1.4)	0	157 (1.2)	57 (3.2)	<0.001
Post-bypass catecholamine	5539 (38)	0	4690 (36)	849 (48)	<0.001
Post-bypass norepinephrine	1849 (13)	0	1693 (13)	156 (8.8)	<0.001
Post-bypass norepinephrine and dobutamine	1229 (8.3)	0	1000 (7.7)	229 (13)	<0.001
Postoperative data					
MV duration (h)	6 (5, 10)	893	6 (5, 9)	8 (6, 21)	<0.001
Catecholamine duration (h)	26 (10, 57)	7673	24 (10, 53)	39 (19, 75)	<0.001
Blood loss at 24 h (mL)	540 (380, 750)	3794	545 (390, 750)	480 (320, 700)	<0.001
Total blood loss (mL)	680 (480, 950)	3434	680 (490, 950)	620 (410, 892)	<0.001
Reintervention	903 (6.1)	0	801 (6.2)	102 (5.7)	0.495
Surgical site infection	540 (3.7)	0	486 (3.7)	54 (3.0)	0.143
Superficial SSI	389 (2.6)	0	356 (2.7)	33 (1.9)	0.030
Deep SSI	151 (1.0)	0	130 (1.0)	21 (1.2)	0.471
ICU LOS (days)	3 (2, 5)	734	3 (2, 5)	4 (2, 7)	<0.001
Total LOS (days)	11 (8, 16)	281	11 (8, 16)	14 (9, 23)	<0.001
D28 mortality	589 (4.0)	0	461 (3.5)	128 (7.2)	<0.001
D90 mortality	802 (5.4)	0	624 (4.8)	178 (10)	<0.001

Continuous variables are expressed as the median and interquartile range (IQR) and were compared using the Mann–Whitney U test. Categorical variables are expressed as *n* (%) and were compared with Fisher’s exact test. BMI, body mass index; CA, cephalosporin antibiotic prophylaxis; CABG, coronary artery bypass graft; CPB, cardiopulmonary bypass; COPD, chronic obstructive pulmonary disease; ESRD, end-stage renal disease; ICU, intensive care unit; LOS, length of stay; MV, mechanical ventilation; SSI, surgical site infection; VGA, vancomycin gentamicin antibiotic prophylaxis.

**Table 2 antibiotics-12-00085-t002:** Multivariate analysis of risk factors for SSI.

	Estimates	CI	*p*
Male	0.50	0.41–0.61	<0.001
Age (years)			
1st quartile <56 y	ref	ref	-
2nd quartile (56, 65) y	0.80	0.60–1.07	0.133
3rd quartile (66–74) y	1.02	0.77–1.34	0.901
4th quartile >74 y	1.27	0.96–1.69	0.095
Obesity	2.22	1.84–2.69	<0.001
Insulin-dependentdiabetes	3.43	2.69–4.35	<0.001
Noninsulin-dependentdiabetes	1.90	1.52–2.38	<0.001
Dyslipidaemia	0.83	0.68–1.02	0.070
Chronic peripheral arterial insufficiency	1.66	1.33–2.06	<0.001
Cardiac insufficiency	1.15	0.75–1.71	0.490
ESRD	1.97	1.10–3.35	0.016
COPD	1.55	1.17–2.03	0.002
Acute infective endocarditis	0.59	0.24–1.22	0.192
Prior cardiac surgery	1.51	0.97–2.30	0.059
CPB time (/10 min)	1.00	0.97–1.03	0.929
CABG	2.68	1.83–3.88	<0.001
Bimammary artery bypass	1.97	1.44–2.76	<0.001
Cardiac transplantation	3.49	1.59–7.15	0.001
Post-bypass norepinephrine and dobutamine	1.71	1.22–2.35	0.001
VGA	0.94	0.68–1.28	0.714

CABG, coronary artery bypass graft; CB, cardiopulmonary bypass; COPD, chronic obstructive pulmonary disease; ESRD, end-stage renal disease; SSI, surgical site infection, VGA, vancomycin gentamicin antibiotic prophylaxis.

**Table 3 antibiotics-12-00085-t003:** Bacterial characteristics of SSI according to antibiotic prophylaxis.

	Overall, *n* = 540	CA, *n* = 486 (90%)	VGA *n* = 54 (10%)	*p* Value
Gram-negative bacteria	197 (36)	169 (35)	28 (52)	0.013
Enterobacterales	175 (32)	151 (31)	24 (44)	0.046
*Escherichia coli*	57 (11)	51 (10)	6 (11)	0.889
*Serratia* spp.	16 (3.0)	15 (3.1)	1 (1.9)	>0.999
*Enterobacter* spp.	43 (8.0)	39 (8.0)	4 (7.4)	>0.999
*Morganella* spp.	14 (2.6)	13 (2.7)	1 (1.9)	>0.999
*Citrobacter* spp.	13 (2.4)	11 (2.3)	2 (3.7)	0.380
*Proteus* spp.	21 (3.9)	18 (3.7)	3 (5.6)	0.456
*Klebsiella* spp.	32 (5.9)	24 (4.9)	8 (15)	0.009
Other (Gram negative) bacteria	7 (1.3)	4 (0.8)	3 (5.6)	0.025
Gram-positive bacteria	361 (67)	334 (69)	27 (50)	0.006
*Staphylococcus* spp.	298 (55)	272 (56)	26 (48)	0.273
*Staphylococcus aureus*	98 (18)	91 (19)	7 (13)	0.297
MSSA	89 (16)	83 (17)	6 (11)	0.262
MRSA	9 (1.7)	8 (1.6)	1 (1.9)	>0.999
CoNS	205 (38)	185 (38)	20 (37)	0.883
*Streptococcus* spp.	8 (1.5)	8 (1.6)	0 (0)	>0.999
*Enterococcus* spp.	62 (11)	61 (13)	1 (1.9)	0.019
Other (Gram positive) bacteria	13 (2.4)	13 (2.7)	0 (0)	0.630
Anaerobic bacteria	18 (3.3)	17 (3.5)	1 (1.9)	>0.999
Fungi	7 (1.3)	4 (0.8)	3 (5.6)	0.025
Associated bacteraemia	129 (24)	117 (24)	12 (22)	0.762

CA, cephalosporin antibiotic prophylaxis; CoNS, coagulase-negative staphylococci; MMSA, methicillin susceptible *Staphylococcus aureus*; MRSA, methicillin resistant *Staphylococcus aureus*; SSI, surgical site infection; VGA, vancomycin gentamicin antibiotic prophylaxis.

**Table 4 antibiotics-12-00085-t004:** Multivariate analysis of risk factors for SSI in a propensity score analysis.

	Estimates	CI	*p*
Male	0.69	0.55–0.85	0.001
Age (years)			
1st quartile < 56 y	Ref	ref	-
2nd quartile [56, 65] y	0.67	0.51–0.88	0.004
3rd quartile [66–74] y	0.86	0.66–1.13	0.269
4th quartile > 74 y	0.77	0.56–1.04	0.086
Obesity	3.47	2.84–4.23	<0.001
Insulin-dependentdiabetes	2.79	2.12–3.69	<0.001
Noninsulin-dependentdiabetes	1.62	1.26–2.07	<0.001
Dyslipidaemia	0.43	0.35–0.54	<0.001
Chronic peripheral arterial insufficiency	1.33	1.01–1.75	0.044
Cardiac insufficiency	1.41	0.92–2.15	0.116
ESRD	4.51	3.21–6.35	<0.001
COPB	1.49	1.14–1.96	0.004
Acute infective endocarditis	1.80	1.36–2.37	<0.001
Prior cardiac surgery	1.54	1.13–2.11	0.006
CB time (/10 min)	0.96	0.94–0.98	0.001
CABG	2.70	1.83–3.97	<0.001
Bimammary artery bypass	1.87	1.28–2.73	0.001
Cardiac transplantation	1.43	0.80–2.58	0.228
Post-bypass norepinephrine and dobutamine	1.62	1.26–2.09	<0.001
VGA	0.89	0.65–1.21	0.457

CABG, coronary artery bypass graft; CB, cardiopulmonary bypass; COPD, chronic obstructive pulmonary disease; ESRD, end-stage renal disease; SSI, surgical site infection; VGA, vancomycin gentamicin antibiotic prophylaxis.

**Table 5 antibiotics-12-00085-t005:** Multivariate analysis of risk factors for SSI in a sensitivity analysis, excluding endocarditis.

	Estimates	CI	*p*
Male	0.52	0.42–0.64	<0.001
Age (years)			
1st quartile < 56 y	ref	ref	-
2nd quartile [56, 65] y	0.81	0.60–1.08	0.155
3rd quartile [66–74] y	1.02	0.77–1.35	0.868
4th quartile > 74 y	1.30	0.98–1.73	0.069
Obesity	2.17	1.79–2.63	<0.001
Insulin-dependentdiabetes	3.52	2.76–4.48	<0.001
Non insulin-dependentdiabetes	1.90	1.52–2.38	<0.001
Dyslipidaemia	0.84	0.68–1.03	0.102
Chronic peripheral arterial insufficiency	1.68	1.35–2.09	<0.001
Cardiac insufficiency	1.12	0.74–1.69	0.606
ESRD	1.75	0.97–3.15	0.061
COPB	1.57	1.19–2.07	0.001
Prior cardiac surgery	1.52	0.98–2.36	0.063
CB time (/10 min)	1.00	0.97–1.04	0.766
CABG	2.65	1.81–3.88	<0.001
Bimammary artery bypass	1.99	1.44–2.77	<0.001
Cardiac transplantation	3.52	1.66–7.45	0.001
Post-bypass norepinephrine and dobutamine	1.66	1.19–2.32	0.003
VGA	1.03	0.76–1.41	0.838

CABG, coronary artery bypass graft; CB, cardiopulmonary bypass; COPD, chronic obstructive pulmonary disease; ESRD, end-stage renal disease; SSI, surgical site infection; VGA, vancomycin gentamicin antibiotic prophylaxis.

**Table 6 antibiotics-12-00085-t006:** Multivariate analysis of risk factors for SSI in a sensitivity analysis restricted to CABG.

	Estimates	CI	*p*
Male	0.36	0.28–0.47	<0.001
Age (years)			
1st quartile < 56 y	ref	ref	-
2nd quartile [56, 65] y	0.80	0.56–1.14	0.218
3rd quartile [66–74] y	0.98	0.69–1.38	0.898
4th quartile > 74 y	1.36	0.95–1.94	0.089
Obesity	2.19	1.73–2.77	<0.001
Insulin-dependentdiabetes	3.59	2.68–4.79	<0.001
Non-insulin-dependentdiabetes	2.21	1.68–2.90	<0.001
Dyslipidaemia	0.94	0.72–1.23	0.655
Chronic peripheral arterial insufficiency	2.02	1.58–2.59	<0.001
Cardiac insufficiency	1.22	0.76–1.95	0.404
ESRD	1.98	1.00–3.92	0.051
COPB	1.42	1.00–2.02	0.052
Prior cardiac surgery	0.25	0.03–1.89	0.177
CB time (/10 min)	0.99	0.93–1.05	0.806
Post-bypass norepinephrine and dobutamine	2.39	1.56–3.65	<0.001
VGA	1.27	0.84–1.92	0.254

CABG, coronary artery bypass graft; CB, cardiopulmonary bypass; COPD, chronic obstructive pulmonary disease; ESRD, end-stage renal disease; SSI, surgical site infection; VGA, vancomycin gentamicin antibiotic prophylaxis.

## Data Availability

Data are available upon request.

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
