# Peer review of "Impact of Antibiotic Prophylaxis on Surgical Site Infections in Cardiac Surgery"

_antibiotics, 2023, doi:10.3390/antibiotics12010085_

Round 1

Reviewer 1 Report

The author performed a very good and elaborate the manuscript very well. I can simply say accept in current form.

Author Response

We thank the reviewer for her/his kind words.

Reviewer 2 Report

Thank you for the opportunity to review this manuscript.

In this manuscript, the authors performed a retrospective study comparing the use of cephalosporins vs vancomycin and gentamycin (VGA) for infection prophylaxis in cardiac surgery. The study includes 14,960 patients in total and during the study period cephalosporins were used as the primary treatment for infection prophylaxis and VGA as second line infection prophylaxis. The authors identified similar rates of infections between the two groups.

Major Comments

1.     The major limitation of this study is the inclusion and exclusion criteria. Patients with ongoing infections seemed to be included in this study, including patients with active endocarditis. In a study comparing antibiotic prophylaxis in cardiac surgery, one would expect that active infections at the time of surgery would be an exclusion criteria given an infection already exists, and the patient is more than likely receiving or has already completed antibiotic therapy at the time of surgery. Both active infection and antibiotic use are significant confounding variables.

2.     The authors state that they utilized a “full matching method” which retained all the patients in their study. While this may be an approach to addressing confounding variables, there are several confounders that are significant and need to be specifically addressed. The authors note that at their center, the internal mammary arteries are almost exclusively used for CABG. While known to have some benefits, one of the major risks associated with bilateral mammary use is sternal wound infection, especially among diabetic patients. Patients in this study receiving cephalosporins were significantly more likely to undergo CABG, receive bilateral mammary artery grafting, and to have diabetes. Additionally, infective endocarditis was 10x more common in the VGA group. These clear differences in risk factors for infection and active major infection limit the ability to compare the groups without excluding some of these populations, or at least ensuring even numbers are present in each group. Whether this be done for the overall group in the propensity matching or a sub group analysis, these and other major contributors to the primary outcome need to be addressed. In a comparison of antibiotic therapy for prophylaxis, it is important to compare patients with similar risk for infection, otherwise these results are limited by numerous significant confounders and it is difficult to draw a conclusion from them.

Minor Comments

1.     In the Patient Management section, the authors state that CABG at their center is performed almost exclusively with internal mammary arteries. This requires clarification. How are patients with multivessel disease treated? In patients with 3 vessel disease, are bilateral mammary arteries used with another conduit such as a radial artery, if vein grafts are not used? If not, are third lesions addressed with PCI, or not at all? If patients regularly receive incomplete revascularization with CABG and no other intervention this should be stated, but if patients are receiving complete revascularization with methods other than vein grafts (arterial grafts or hybrid procedure with PCI), this should be stated.

2.     The tables and figures provided are appropriate and complement the information provided in this study. 

Reviewer 3 Report

In the manuscript ID: antibiotics-2090933 the authors report a retrospective study about the efficiency of cephalosporins or vancomycin/gentamicin treatments in preventing surgical site infections. By analysing data recovered from almost 15000 patients, they concluded the vancomycin/gentamicin association (VGA) could represent a suitable option to treat subjects showing allergies to β-lactams or presenting complicated clinical features. As particular observation, they reported a different infection etiology dependent on the selected treatment, thus highlighting the need for the development of a broad-spectrum therapy to counteract the insurgence of critical surgical site infections, especially in patients subjected to cardiac surgery.

The manuscript is interesting and scientifically sounding. The introduction, although brief, provides the necessary information to understand the topic, the methods are well described and detailed, the results are well presented, supported by a strong statistical analysis and well commented in the discussion section.

The main concern of the study, i.e., the discrepancy in the percentages of the patients treated with cephalosporins or with VGA, was well solved by statistical analysis and is consistent with data reported in literature. Did the authors consider comparing the VGA-treated group with an equal subset of subjects treated with cephalosporins and presenting similar clinical features? Please comment.

The Gram-negative etiology of infections in VGA-treated group is not surprising, as vancomycin cannot be applied to these bacteria and gentamycin resistance is widely spread in the nosocomial environment in Gram-negative species (Fleischmann et al., 2020 doi: 10.1128/AAC.01711-19; Solomon et al., 2021 doi: 10.1371/journal.pone.0255410). This should be underlined in the manuscript. The suggestion of vancomycin/cefazolin combinations is interesting.

Based on these observation, minor revisions are recommended before publication in “Antibiotics”.

MINOR COMMENTS

Please explicit the acronyms (e.g., COPD and ESRD) even in the text, not only in the tables captions;

Line 338, please delete “was made”.

Author Response

Please see the attachtment

Round 2

Reviewer 2 Report

Thank you for the opportunity to review this manuscript.

In this manuscript, the authors performed a retrospective study comparing the use of cephalosporins vs vancomycin and gentamycin (VGA) for infection prophylaxis in cardiac surgery. The study includes 14,960 patients in total and during the study period cephalosporins were used as the primary treatment for infection prophylaxis and VGA as second line infection prophylaxis. The authors identified similar rates of infections between the two groups.

Comments

1.     The authors have addressed my previous comments. I have no further suggestions for improvement at this time.